# CpG ODN D35 improves the response to abbreviated low-dose pentavalent antimonial treatment in non-human primate model of cutaneous leishmaniasis

**Seth G Thacker**[1], **Ian L. McWilliams**[1], **Beatrice Bonnet**[2], **Lydia Halie**[1], **Serge Beaucage**[3], **Swaksha Rachuri**[1], **Ranadhir Dey**[4], **Robert Duncan**[4], **Farrokh Modabber**[2], **Stephen Robinson**[2], **Graeme Bilbe**[2], **Byron Arana**[2]*, **Daniela Verthelyi**[1]*

**1** Laboratory of Immunology, Office of Biotechnology Products, Center for Drug Evaluation and Research, Food and Drug Administration, Silver Spring, Maryland, United States of America, **2** Drugs for Neglected Diseases initiative (DNDi), Geneva, Switzerland, **3** Laboratory of Biological Chemistry; Office of Biotechnology Products, Center for Drug Evaluation and Research, Food and Drug Administration, Silver Spring, Maryland, United States of America, **4** Laboratory of Emerging Pathogens, Division of Emerging and Transfusion Transmitted Diseases, Office of Blood Research and Review, Center for Biologics Evaluation and Research, Food and Drug Administration, Silver Spring, Maryland, United States of America

* barana@dndi.org (BA); Daniela.Verthelyi@fda.hhs.gov (DV)

**Data Availability Statement:** All relevant data are within the manuscript and its Supporting Information files.

## Abstract

Cutaneous leishmaniasis (CL) affects the lives of 0.7–1 million people every year causing lesions that take months to heal. These lesions can result in disfiguring scars with psychological, social and economic consequences. Antimonials are the first line of therapy for CL, however the treatment is lengthy and linked to significant toxicities; further, its efficacy is variable and resistant parasites are emerging. Shorter or lower dose antimonial treatment regimens, which would decrease the risk of adverse events and improve patient compliance, have shown reduced efficacy and further increase the risk emergence of antimonial-resistant strains. The progression of lesions in CL is partly determined by the immune response it elicits, and previous studies showed that administration of immunomodulatory type D CpG ODNs, magnifies the immune response to *Leishmania* and reduces lesion severity in nonhuman primates (NHP) challenged with *Leishmania major* or *Leishmania amazonensis*. Here we explored whether the addition of a single dose of immunomodulating CpG ODN D35 augments the efficacy of a short-course, low-dose pentavalent antimonial treatment regimen. Results show that macaques treated with D35 plus 5mg/kg sodium stibogluconate (Sb$^V$) for 10 days had smaller lesions and reduced time to re-epithelization after infection with *Leishmania major*. No toxicities were evident during the studies, even at doses of D35 10 times higher than those used in treatment. Critically, pentavalent antimonial treatment did not modify the ability of D35 to induce type I IFNs. The findings support the efficacy of D35 as adjuvant therapy for shorter, low dose pentavalent antimonial treatment.

**Funding:** DNDi received a grant from the Global Health Innovative Technology Fund (https://www.ghitfund.org/) (G2015-211) that contributed to the performance of these studies. The funders had no role in study design, data collection and analysis, decision to publish, or preparation of the manuscript.

**Competing interests:** Daniela Verthelyi is named as an inventor on a patent covering D35 filed on April, 1999, US 7,960,356 B2 under the Serial number 60/128,898. WO2000/061151. The patent is owned by the US government and licensed to DNDi.

## Author summary

Cutaneous leishmaniasis is a devastating disease that affects close to a million people every year. Its clinical presentation ranges from small uncomplicated lesions that heal over a few months to debilitating large chronic or recurring lesions that result in disfigurement, stigma, and economic loss. Antimonials are the first line treatment for cutaneous leishmaniasis in most countries, but the lengthy treatment schedules, significant associated toxicities, and the emergence of resistant strains, require the development of alternative strategies. As the immune response is a key determinant of disease course, immunomodulatory therapies could be harnessed to act in concert with antimonials to improve the safety and efficacy of CL treatment. Synthetic oligonucleotide D35 selectively activates plasmacytoid dendritic cells and was previously shown to reduce the severity of *L. major* and *L. amazonensis* lesions in rhesus macaques, but its activity in combination with antimonials was unknown. Our studies show that a single subcutaneous dose of innate immune modulator D35 improved the response to a low-dose abbreviated antimonial course, reducing the severity of the lesions and accelerating healing in primates. No toxicities were evident with D35 at doses ten-fold higher than the effective dose. The studies suggest that the combined therapy strategy shows clinical promise.

## Introduction

Cutaneous leishmaniasis (CL) is a zoonotic, vector-borne parasitic disease that affects 0.7–1 million mostly young patients every year [1, 2]. Despite the high incidence rate and vast geographic expansion, spreading throughout the Mediterranean, South American and Middle Eastern countries [1, 2], CL remains a neglected tropical disease with few effective intervention strategies [3]. Clinically, CL usually presents as small papules at the site of infection that may progress to form nodules and then open sores with raised borders and central ulcers that can be covered with scales or crust. The lesions are usually painless but can be painful, particularly if superinfected with bacteria. While most lesions heal within 18 months, they can result in disfiguring scars that lead to life-long social stigma and economic loss [3–5]. Depending on the parasite strain and the immune response it elicits, CL can also take the form of diffuse cutaneous leishmaniasis, disseminated cutaneous leishmaniasis, Leishmania recidivans, or adopt the mutilating mucocutaneous form, which is harder to control [1–3].

Current treatment options for leishmaniasis include pentavalent antimonials (Sb$^V$: sodium stibogluconate or meglumine antimoniate), amphotericin, miltefosine, and pentamidine. However, due to availability, cost, and relative safety and efficacy, Sb$^V$ developed in the 1930's, remains the primary drug employed against CL [3, 6]. In several studies, treatment with Sb$^V$ accelerated healing of CL lesions when used at 10–30 mg/kg/day IV or IM for 20–30 days, but the success rate ranges between 25 and 90% depending on the population and the strain of *Leishmania*, and even at the higher doses there are relapses [3, 7]. Importantly, this prolonged high-dose treatment regimen is associated with severe adverse effects including cardiotoxicity (prolonged QTc interval, premature beats, tachycardia, fibrillation), pancreatitis, liver and kidney toxicities, malaise, myalgia, and anorexia [8, 9]. Lastly, there are growing reports in India, Iran, and Peru of emerging resistance to antimonials in the field [10–14]. Several studies have explored reducing the dose or shortening the course of antimonial treatment to improve compliance and reduce toxicities with limited success [15, 16].

It is well established that the clinical outcome in CL is determined by the type of parasite and the immune response of the host [17, 18]. In subjects who produce antibodies to the

parasite but do not mount adequate cellular immune responses, CL can evolve into a dissemi-nated form of the disease with multiple large lesions [3]. Conversely, in patients that mount strong Th1 responses CL can present with an aggressive form called mucosal leishmaniasis [18]. Thus, interventions to modulate the immune response to the parasite and improve the disease outcome must negotiate a fine balance in which cellular responses are enhanced with-out leading to excessive inflammation or excessive cytolytic responses [3, 18]. To date, most immunomodulatory treatments have centered around topical applications of innate immune response modifiers, such as imiquimod (TLR7/8 agonist) or GM-CSF to aide lesion healing, albeit with mixed results [19–22]. However, studies that combined heat-killed *Leishmania* pro-mastigotes and bacille Calmette-Guérin (BCG) with low dose antimonials for patients with CL or PKDL suggested that the addition of immune modulators could achieve comparable effi-cacy to full dose antimonials with fewer adverse effects [23, 24]. Lastly, there are some studies suggesting that imiquimod induces the activation of dendritic cells and the production of type I interferons, improving the efficacy of Glucantime therapy in *L. major* patients, although topi-cal imiquimod can induce psoriatic-like lesions [19, 25]. Together these studies suggest that the addition of an immune response modulator may allow for shorter treatment courses, reducing toxicities and lowering the risk of the development of resistance; however, a safe and effective regimen has yet to be identified [23].

Rhesus macaques are a useful model for testing therapies for CL as intradermal challenges with metacyclic promastigotes induce the formation of a lesion that recapitulates the evolution of the lesions in patients. In this model, 3–4 week regimens of antimonials at 20mg/kg/d reduce the severity of the CL lesions, but courses with reduced or abbreviated therapies show minimal or transient therapeutic effect [26]. We have previously shown that treatment with type D CpG ODN improves the outcome of *L. amazonensis* or *L. major* infections in macaques. Administration of type D CpG ODN 3 days before or up to 15 days post-infection can reduce the lesion severity regardless of route (ID, IM or SC) suggesting that the product induces a systemic immunomodulatory effect [27–29]. Importantly, even with the reduction in lesion size and accelerated healing, the animals developed long-lasting protective memory responses to the parasite [29]. Lastly, the improved response was evident even in macaques with simian AIDS, an important concern given the overlap in endemic areas [30].

Type D CpG ODNs are short synthetic oligonucleotide sequences that have a single PuPyCpGPuPy motif surrounded by self-complementary bases [31]. They selectively stimulate TLR9-bearing human and non-human primate plasmacytoid DC (pDC) to secrete IFNα [32], which in turn leads monocytes to mature into functionally active DC [33], and NK cells to secrete IFNγ [31]. Of note, D type ODNs have a 3′ end poly(G) motif, that can self-associate via Hoogsteen base-pairing to form parallel quadruplex structures called G-tetrads. The forma-tion of multimers may contribute to the ODN's localization to early endosomes where it sig-nals through TLR-9, however the formation of tetrads poses formidable challenges for the synthesis, purification, and characterization of CpG ODN type D and hinders its clinical devel-opment. Further, previous studies showed that different TLR ligands can synergize to augment their immunomodulatory effects even when present at very low levels [34, 35], and it was therefore possible that impurities in the research-grade D35 preparation used in previous stud-ies could have contributed to its immunomodulatory and clinical effect. In these studies, we assess whether D35, manufactured under ultrapure conditions and formulated to avoid the formation of aggregates, induces an immunomodulatory effect that can be used to improve the response to a suboptimal regimen of $Sb^V$. We demonstrate that administration of a single dose of highly purified D35 (1 mg/kg SC) in macaques challenged with *L. major*, prior to a short, low-dose antimonial ($Sb^V$) regimen, reduced lesion size, and lowered time to re-epitheli-zation compared to untreated or $Sb^V$ only treated animals. Importantly, macaques receiving

repeated and escalating doses of D35, that were over 10-fold higher than the effective dose, showed no toxicities. Together these results suggest that administration of D35 can safely enhance the efficacy of a therapeutic regimen with lower exposure to $Sb^V$.

## Materials and methods

### Oligonucleotides

Gene Design, Inc (Japan) was contracted to synthesis a batch of ultrapure D35 (GGtgcatcgatg-caggggGG, phosphorothioate bases in uppercase and phosphodiester bases in lowercase). D35 was supplied formulated with 5% maltose. Purity of D35 lots was checked by HPLC and PAGE gel, and endotoxin levels were found to be < 0.5 EU/mg by Endosafe PTS (Charles River).

### Mononuclear cell isolation and stimulation

Buffy coats from healthy human blood donors were obtained from the National Institutes of Health Department of Transfusion Medicine. Mononuclear cells were isolated by density gradient centrifugation of PBMCs over Ficoll-Hypaque as described [36]. Cells were washed three times and cultured in RPMI 1640 supplemented with 10% heat-inactivated FCS, 1.5 mM l-glutamine, and 100 U/ml penicillin/streptomycin. In a 24 well multiplate, $5 \times 10^6$ cells/well were cultured in the presence of 1 μM ODN D35. Where indicated, $Sb^V$ (Pentostam-Sodium Stibogluconate, CDC) was added at the same time as D35. RNA was collected after 24 h of stimulation while supernatants were collected after 72 h and stored at -20˚C until further analysis.

### CAL-1 cell culture and stimulation

CAL-1 cells, a pDC cell line [37], were provided by Dr. T. Maeda (Department of Island Medicine, Nagasaki University, Japan). Cells were cultured in RPMI supplement with 10% FBS, 1.5 mM l-glutamine, 100 U/ml penicillin/streptomycin, 1mM Na Pyruvate, 10 mM Hepes, and non-essential amino acids (ThermoFisher Scientific). Cells were supplemented with fresh media every 2–3 days and the concentration was kept between $0.3–1.5 \times 10^6$ cells/ml. Cells were stimulated in a 48 well multiplate (ThermoFisher Scientific) at the final concentration of $1.6 \times 10^6$ cells/ml. Cells were stimulated for 18 hours with D35. This time point was found to be optimal for assessing gene induction due to D35 stimulation. Media was removed, and RNA was isolated from the cells.

### Nucleic acid analysis

RNA was isolated from cells by the addition of Trizol (ThermoFisher Scientific, Fremont, CA) following the manufacturer's recommendations. Following RNA isolation, cDNA was synthesized using 1000 ng total RNA and MultiScribe reverse transcriptase (ThermoFisher Scientific). Gene expression was measured using standard qPCR, low density taqman array (human immunology panel) or by NanoString (NanoString, Seattle, WA). Gene expression in qPCR experiments were quantified using the ddCT method, and in NHP studies each animal's prestudy measurements were used as the reference. Gene expression by NanoString was determined using the nSolver v3.0 software and the advanced analysis module v1.1.5.

### Cytokine quantification

Cytokine levels were analyzed by cytometric bead array. For human samples IFNα, IFNγ, CXCL10, and IL6 were measured (ThermoFisher Scientific). For macaque samples, IFNγ, IL8, IL10, IL15, and IL17 were measured using a custom kit from Millipore Sigma (Burlington,

MA). The manufacturer's recommended protocol was followed, and plates were read on Luminex 200 system (MilliporeSigma).

## Primate use, ethics statement

These studies were carried out in strict accordance with the recommendations in the Guide for the Care and Use of Laboratory Animals of the National Institutes of Health and using protocols 2016–17 and 2017–62 approved by the Food and Drug's Animal Care and Use Committee (ACUC) and conducted in accordance with the Association for Assessment and Accreditation of Laboratory Animal Care's (AAALAC) guidelines. The animals were housed in approved facilities and monitored daily by veterinarians and facility personnel. The macaques were kept in double housing (paired), fed a complete diet that included fresh fruits and vegetables. The cages were arranged in large rooms to allow the monkeys visual, olfactory and auditory interactions with each other. Food and water were available ad libitum and vitamins were provided. The animals were also provided with environmental enrichment, such as toys designed especially for monkeys, to promote psychological well-being. All inoculations, measurements and biopsies were performed under anesthesia using Ketaject; 5–7 mg/kg of body weight (Phoenix Pharmaceuticals), anesthesia was reversed with atipamezole hydrochloride (Antisedan, 100 μg/kg of body weight; Zoetis Services).

## D35 dose escalation study

Two male and two female cynomolgus macaques (*Macaca fascicularis*, 6-11yrs old; weights 4.3 kg to 12 kg), were housed at the National Institutes of Health Animal Center. Macaques were anesthetized with ketamine prior to all procedures (Ketaject; 5–7 mg/kg of body weight; Phoenix Pharmaceuticals), following the procedure the anesthetic was reversed with atipamezole hydrochloride (Antisedan, 100 μg/kg of body weight; Zoetis Services). Blood was collected via femoral vein and following blood collection animals were injected with D35 in the chest (SC). Twenty-four hours after D35 administration, animals were anesthetized again and 5 mm skin biopsies (Miltex, Inc) were taken at the site of D35 administration or on the contralateral side of the chest. D35 injections were performed on alternating sides of the thorax and at least 2 inches away from a previous biopsy site to minimize the confounding effect of any residual inflammation due to a previous injection or biopsy. For a visual representation see S1 Fig. Skin biopsies were placed immediately into Trizol and kept on ice until samples could be processed. A rectal temperature was taken before D35 administration and 24 h after D35 administration when the animals were anesthetized. One week after D35 administration macaques were again anesthetized and blood was collected via the femoral vein. Animals were allowed to recover for an additional 14 days before the next dose of D35 was administered.

## *In vivo* interaction between D35 and Sb$^V$

Male rhesus macaques (*Macaca mulatta*) were obtained from the National Institutes of Health (NIH) colony in South Carolina [38], housed in approved facilities, and monitored daily by veterinarians and facility personnel. Animals were 2 years of age, and their average weight was 3.4 kg (range 2.6-4kg). Treatment groups were balanced for weight prior to starting the study. Animals were grouped into five groups based on the following treatments, Saline, Sb$^V$ 20 mg/ml, Sb$^V$ 5 mg/ml, Sb$^V$ 2.5 mg/kg, Sb$^V$ 0 mg/ml. Following 3 doses of Sb$^V$ given every other day IM, the animals in groups Sb$^V$ 20, 5, 2.5, 0 mg/kg were given a single dose of D35 1 mg/kg SC at a distant site. Blood and 4 mm skin biopsies (Miltex, Inc) were taken distant from the sites were drugs were previously administered. Skin biopsies were placed immediately into Trizol and kept on ice until samples could be processed. Macaques were anesthetized with ketamine

for all procedures (Ketaject; 5–10 mg/kg of body weight; Phoenix Pharmaceuticals), following the procedure the anesthetic was reversed with atipamezole hydrochloride (Antisedan, 100 μg/kg of body weight; Zoetis Services). The animals showed no sign of itching or pain related to the lesions or the biopsies and efforts were made to minimize their suffering.

## The impact of D35 on *L. major* infection

Male and female rhesus macaques (*M. mulatta*) were obtained from the NIH colony in South Carolina [38] and housed in FDA AAALAC accredited facilities. Animals ranged in age from 2 to 6 years, and the average weight was 4.2 kg (range 2.6–9.62 kg). Treatment groups were balanced for age and gender and assigned prior to starting the study. Macaques (n = 3-4/group) were challenged intra-dermally (ID) with 3 x10$^6$ metacyclic promastigotes on the forehead at three separate sites (1 on the right and 2 on the left). The inoculation on the right remained untouched until the resolution of the lesions and used for lesion measurements, while the two on the left were used for biopsies at 11- and 22- days post infection (DPI). Lesion size was monitored by a blinded researcher measuring an untouched lesion (in mm) and by digital photography every 2–3 days for the first 4 weeks, followed by weekly measurement from then on. When 70% of lesions measured greater than 3 x 3mm, half the macaques were treated with D35 (1 mg/kg SC) or saline, in the hind quarters. Three days following D35 administration, the course of daily administration of Sb$^V$ (5 mg/kg/d for 10 days, IM in the thigh) was initiated. This resulted in four treatment groups for the animals defined as group 1-Saline, group 2-Sb$^V$, group 3-D35, and group 4-D35 + Sb$^V$. Graphical representation of study layout can be found in S2 Fig. Blood, serum samples and 4 mm diameter punch biopsies (Miltex, Inc) of the forehead skin were obtained at 11- and 22 DPI. Skin biopsies were divided into three pieces; two were immediately frozen in liquid nitrogen while the third was submerged in TRIzol and stored at -80˚C until processing. Macaques were anesthetized with ketamine for all procedures (Ketaject; 5–10 mg/kg of body weight; Phoenix Pharmaceuticals), following the procedure the anesthetic was reversed with atipamezole hydrochloride (Antisedan, 100 μg/kg of body weight; Zoetis Services). All experiments were approved by the Animal Care and Use Committee (ACUC), conducted in accordance with the Association for Assessment and Accreditation of Laboratory Animal Care's (AAALAC) guidelines, the animals were housed in approved facilities and monitored daily by veterinarians and facility personnel.

*L. major* clone VI promastigotes (MHOM/IL/80/Friedlin provided and prepared by Dr David Sacks, NIH) were maintained as follows: promastigotes were grown at 26˚C in medium 199 (M199) supplemented with 20% heat-inactivated FCS (Gemini Bio-Products), 100 U/ml penicillin, 100 μg/ml streptomycin, 2 mM l-glutamine, 40 mM Hepes, 0.1 mM adenine (in 50 mM Hepes), 5 mg/ml hemin (in 50% triethanolamine), and 1 mg/ml 6-biotin (M199/S). Infective-stage metacyclic promastigotes were isolated from stationary cultures (5–6 d) by density gradient centrifugation and were grown and isolated as described previously [39].

## Tissue homogenization

Samples for RNA extraction were thawed and mixed with 0.5 cm$^3$ of 2.0 mm Zirconia Beads (Biospec). Tissue was homogenized using the Precellys 24, Cryolys system (Bertin Technologies) with the following settings: 3 cycles at 6800rpm, with 30s of alternate run and pause time, program was carried out twice. Homogenate was transferred to a clean 2 mL Eppendorf tube and total RNA was isolated as described above.

## NanoString mRNA profiling

The NanoString nCounter NHP Immunology Panel (NanoString Technologies, Seattle, WA) was used for skin samples, and PBMC expression was analyzed with a custom panel of 85 genes (S1 Table) per manufacturer's instructions. Briefly, probes were hybridized to 100 ng of total RNA for 19 h at 65°C, after which, excess capture and reporter probes were removed, and transcript-specific ternary complexes were immobilized on a streptavidin-coated cartridge. The code set contains a 3′ biotinylated capture probe and a 5′ reporter probe tagged with a fluorescent barcode, two sequence-specific probes for each of 770 transcripts. All solution manipulations were carried out using the NanoString preparation station robotic fluids handling platform. Data collection was carried out with the nCounter Digital Analyzer to count individual fluorescent barcodes and quantify target RNA molecules present in each sample. Normalization was performed based on a standard curve constructed using the spike in exogenous control samples. Background hybridization signal was determined using the spike in negative controls provided. mRNAs with counts lower than the mean background +2 standard deviations were considered to be below the limit of detection. The nSolver (v.3.0) user interface (Nanostring) was used to operate the nCounter advanced analysis module, which employs the R statistical software. The advanced analysis module was used to identify the mRNA transcripts that were significantly elevated above each animal's pre-study levels (p<0.05). The differences in expression level comparing saline and D35 treatment for individual genes were tested by one tailed t test with an alpha of 0.1 (Graph pad Prism 7.0). The raw datasets generated during the current study are available from the corresponding author upon reasonable request.

To better understand the differences in gene expression between treated and untreated macaques, we analyzed the genes that were significantly different between the two groups (1 tailed t test, alpha = 0.1) using Ingenuity Pathway Analysis. Pathways which were disease specific and not relevant to the current study where removed see S2 Table for complete list of enriched pathways.

## Statistics

All statistics were carried out using Prism 7 (Graphpad Software, La Jolla, CA) apart from the NanoString analysis. For analysis of Nanostring data, nSolver™ v3.0 was used. Briefly, gene expression was first normalized to positive and negative controls and then to the geometric mean of housekeeping genes (*ABCF1*, *HPRT1*, *POLR2A*, *SDHA*, *TBP*, *TUBB*). The changes in gene expression were assessed as counts of mRNA relative to each animal's baseline. Hierarchical clustering was carried out using Ward's minimum variance method in R. The determination of which genes were significantly increased over their paired pre-study state was analyzed using the advanced analysis module v1.1.5 of nSolver. T tests (α 0.1) were used to test differences in gene expression between saline and D35 samples. All analyses involving more than two groups were performed by ANOVA followed by Tukey's multiple comparisons test.

## Results

### Characterization of ultrapure type D CpG ODN D35

Type D CpG ODN have been shown to act as innate immune modulators stimulating plasmacytoid dendritic cells to produce type I IFNs, IFN-inducible genes (ISG), and effectively reducing the severity of lesions in macaques infected with *L. major* [40–42]. Their activity has been linked to a single CpG motif encased in a self-complementary core sequence and a 3′ end poly(G) track that favors G-tetrad formation leading to multimerization. Although

multimerization appears necessary for intracellular localization to early endosomes and signaling via Toll-like receptor 9 (TLR-9), it can result in product polymorphisms with an increased risk of uncontrolled aggregation and precipitation during manufacture, thereby hampering their clinical development [31, 43, 44]. To examine whether biological activity would be reduced if the product's multimerization was controlled, a highly purified synthetic oligonucleotide was formulated in 5% maltose to produce an ultrapure lot of CpG ODN D35 (D35) with low levels of impurities and aggregates (S2 Fig). To assess its biological activity, we compared the mRNA expression pattern for 90 immune-related genes in PBMCs stimulated with either research grade D35 or ultrapure D35 (1μM). As shown in Fig 1A and 1B, despite reduced multimerization, both batches of D35 induced a similar gene expression profile in PBMC of healthy blood donors (r = 0.96; p<0.001). However, closer examination of the levels of mRNA induced by each lot suggested that there were subtle differences in the magnitude of the response for some genes (Fig 1B). This was confirmed when we assessed the levels of individual cytokines in supernatants at 72 h. While all lots of D35 induced similar levels of CXCL10, the ultrapure oligos induced significantly lower level of IFNα and IL6 as well as a trend towards lower levels of IFNγ that did not reach statistical significance but could be clinically relevant (Fig 1C). As expected, the response in PBMC is quite variable from donor to donor, therefore we next established a cell line-based assay to reproducibly monitor the bioactivity of D35 using the pDC Cal-1 cells. As shown in Fig 1E, research grade D35 and ultrapure D35 induced similar levels of MX-1 in these cells when used at 0.3–3 μM, however, at higher concentrations of D35 (>6 μM), the induction of MX-1 was more modest for the formulated than the research grade D35. While the lower activity of D ODN that do not form aggregates is consistent with previous reports, these studies suggested that the ultrapure D35 conserves its immunomodulatory activity at concentrations previously shown to be immunostimulatory *in vitro* [28, 30, 40]. Lastly, to establish that both ODN stimulated PBMC from primates, we stimulated human and rhesus macaque PBMC with 0.3 and 1 μM of D35 for 24 h. As shown in Fig 1D, both preparations induced similar mRNA levels of MX-1, confirming that non-human primates are a good model to test the activity of these ODN *in vivo*. In summary, the *in vitro* data suggested that ultrapure D35 induced a similar response to research grade D35 despite the reduced formation of aggregates and reduced impurities, thus all ensuing studies were conducted using the ultrapure D35.

To confirm the activity of the D35 *in vivo*, we next inoculated three cynomolgus macaques in the torso (SC) with increasing doses of D35 at three-week intervals (S1 Fig). A fourth macaque was inoculated every 3 weeks with saline and served as an experimental control. Skin biopsies collected 24 hours post inoculation showed increased expression of interferon stimulated genes (ISGs), *MX1*, *OAS1*, *CXCL10*, and *IRF7*, starting at 1 mg/kg at the site of injection (ipsilateral, local response at the inoculation site) as well as the contralateral (systemic response) side of the torso (Fig 2A–2D). This indicated that D35 modifies the cytokine milieu beyond the inoculation site. At the injection site, the ISG expression levels were higher but appeared to plateau for doses higher than 3 mg/kg. In contrast, the contralateral biopsy showed a dose-dependent increase in mRNA expression levels. The minimal changes in mRNA expression for *IL6* and *IL8* in skin suggest that the pro-inflammatory effect of D35 is mild (Fig 2E & 2F). Of note, skin biopsies taken 3 weeks after the last inoculation with D35 showed that the mRNA levels had not returned to baseline for MX1, OAS1, and CXCL10, suggesting that the systemic immunomodulatory effect of D35 is modest but sustained *in vivo*. The immune activation induced by D35 was also evident in peripheral blood, as treated macaques showed increased mRNA levels for *MX1*, *OAS1*, and *IRF7* at all concentrations tested (Fig 2G–2J). Unlike the skin biopsies, the levels of mRNA expression in PBMC returned to baseline 7 days post-treatment except for macaque C71, which showed relatively higher levels of *MX-1* and

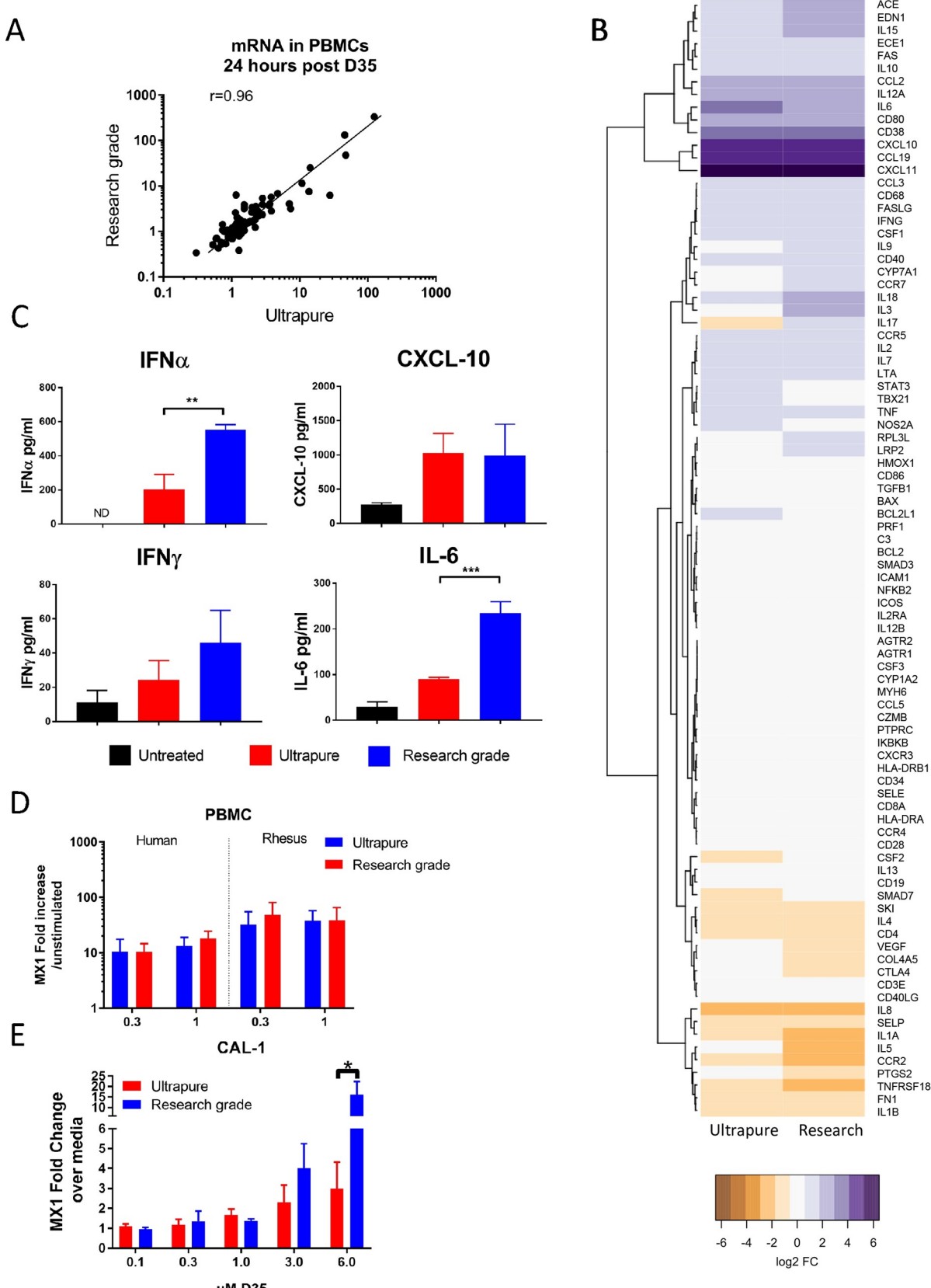

**Fig 1. D35 formulation reduces product aggregates.** PBMCs from healthy donors (n = 3–5; 4 x $10^6$ cells/ml) were stimulated in triplicate in a 24 well plate with ultrapure or research grade D35 (1 μM). The induction of 88 immune related genes by ultrapure and research grade D35 after 24 h of stimulation is shown by scatter plot (A) and heatmap (B). Degree of correlation was calculated in Prizm 7 and gene clustering in the heatmap was calculated by Ward's minimum variance method in R. mRNA for *MX1* shows similar dose-dependent levels of mRNA expression by qPCR in response to *in vitro* stimulation. (C) Cytokine levels in supernatants of PBMCs stimulated for 72 h in the presence of 1 μM D35 (Mean ± SD of 3–5 donors). (D/E) PBMC (D) (4 x $10^6$/ml stimulated in a 24 well plate) or CAL-1(E), a human pDC cell line (3.33 x $10^6$/ml stimulated in a 48 well plate), were incubated with increasing concentrations of D35 as indicated, and expression of *MX1* mRNA, a type I IFN response gene, was used as a surrogate for IFN-α production after 24 h in culture. Data from PBMC were generated using PBMC from 3–5 healthy human donors and 8 adult macaques. CAL-1 experiments were performed at least 3 separate times in triplicate. ** indicates p<0.01 and *** indicates p<0.001. All statistics with human PBMCs were calculated using a paired analysis. CAL-1 data was analyzed by t-test.

*IRF7* mRNA levels in peripheral blood 7 days after receiving the 6, 9, and 12 mg/kg doses. Three weeks after the last dose, the ISG mRNA levels in peripheral blood had returned to baseline. As observed in skin, D35 did not induce detectable increases in *IL6* or *IL8* in PBMC confirming the mild pro-inflammatory effect of D35 (Fig 2K & 2L). Importantly, no fever, changes in CBC, chemistry panel, or weight were evident during the study (S3 Table). Lastly, autopsies of the macaques did not show any pathologic changes. Together these data indicate that the ultrapure D35 has biological activity despite having reduced impurities. Given that D35 induces detectable local and systemic biological activity at 1 mg/kg, which is the dose previously used to treat CL in macaques, this dose was used to explore the use of D35 as an adjunct treatment to reduce the dose of $Sb^V$.

## In vitro interaction of $Sb^V$ and D35

Antimonials are the first line treatment for CL. Their mechanism of action is incompletely understood, but they are thought to act by priming the respiratory burst of phagocytes, increasing the sensitivity of cells to cytokines and interferons, and directly decreasing parasite DNA, RNA, protein synthesis [45–47]. While $Sb^V$ and D35 have distinct mechanisms of action, it was unknown whether $Sb^V$ would alter the immunomodulatory activity of D35 in CL. To explore whether the two treatments could be used in combination, we first determined whether $Sb^V$ modifies the effect of D35 *in vitro*. PBMCs from healthy human blood donors were stimulated with 1μM D35 alone or in the presence of increasing concentrations of $Sb^V$ (20, 100, 500 μg/ml). These doses correspond to the reported peak serum levels in NHP following administration of a 20mg/kg dose (20–40 μg/mL), and supramaximal concentrations of $Sb^V$ (100 and 500 μg/mL) [26, 48–50]. As expected, treatment with D35 induced the expression of IFNα, IFNγ, and CXCL-10 both at the mRNA (24h) and protein levels (72h) (Fig 3). Treating the cells with 20 ng/ml $Sb^V$ did not modify the cytokine levels in these cells (Fig 3). In cells exposed to both, the presence of $Sb^V$ in the culture did not significantly modify the mRNA response to D35 at concentrations consistent with Cmax for $Sb^V$. At higher $Sb^V$ concentrations, however the cells from some donors showed increased levels of mRNA for IFNα and IFNγ, as well as relatively increased levels of CXCL10 (Fig 3A). Further, at 500 μg/ml of $Sb^V$ there was a reduction in the levels of IFNα, CXCL10, and IFNγ induced by D35 in supernatants (Fig 3B) suggesting that exposure to $Sb^V$ could blunt the immune activation of D35, albeit at concentrations over 10-fold higher than expected in patients.

## In vivo effects of $Sb^V$ on D35 treatment

As we had observed an effect of $Sb^V$ on the response to D35 at high doses *in vitro*, we next examined whether $Sb^V$ at clinically relevant doses could affect the *in vivo* response to D35. Rhesus macaques (n = 6/group) received three doses of $Sb^V$ IM (0, 2.5, 5 or 20mg/kg) on 3

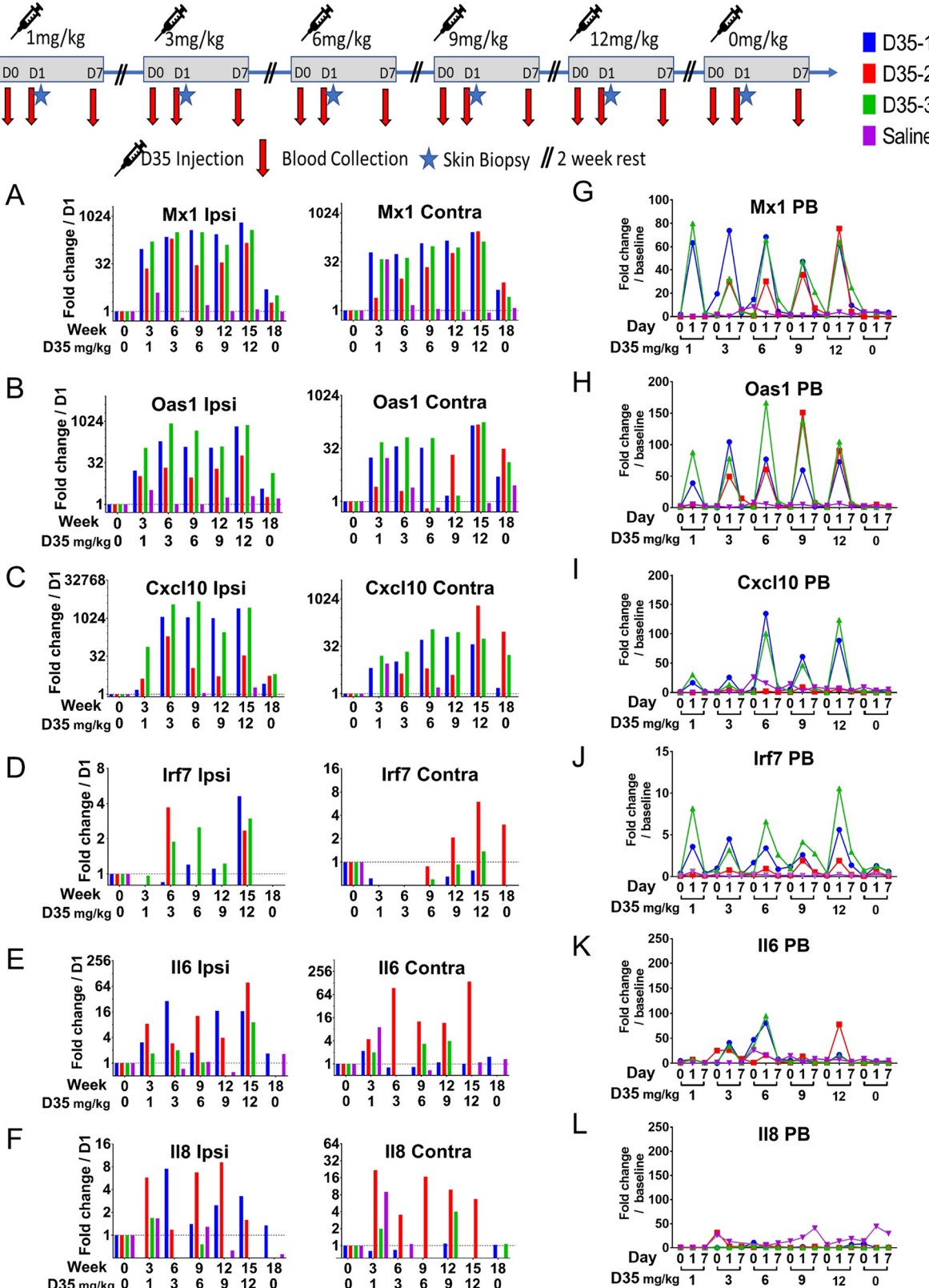

**Fig 2. Local and systemic response to administration of increasing doses of D35 *in vivo*.** Three monkeys received increasing doses of D35 and one received saline The monkeys were injected subcutaneously on alternating sides of the chest with 1, 3, 6, 9, and 12 mg/kg at 3-

week intervals. Blood samples were collected prior to each SC D35 injection, as well as 1- and 7-days post-injection, after which the monkeys were rested for two weeks and then a new cycle started. A final sampling cycle was taken after the last inoculation to assess whether mRNA levels had reverted to baseline. Skin biopsies were collected from the site of injection (ipsilateral) and on the contralateral side of the chest 24 hours post-treatment to assess the local and systemic effect of the treatment on skin. mRNA expression levels were measured by qPCR. (A-F) shows the individual mRNA levels at ipsilateral and contralateral sites. (G-L) Shows gene expression from PBMCs.

consecutive days followed by a single subcutaneous dose of D35 (1mg/kg SC). Saline treated animals were used as controls. The animals did not show any signs of Sb$^V$ drug-related fever, loss of appetite, or muscle tenderness even at the highest dose regimen. As shown in Fig 4A, treatment with D35 induced a detectable increase in mRNA for *MX1*, *CXCL10* and *IRF7* in PBMC (12.1, 5.7, and 2.7-fold respectively, relative to the baseline for each individual macaque). Treatment with Sb$^V$ did not modify the induction of *MX1*, *CXCL10*, or *IRF7* regardless of dose.

Since skin is the target organ for CL, we next determined whether D35 treatment induces a systemic change in gene expression in the skin. Full depth skin biopsies were obtained at a site distant from the administration site of either drug and tested for cytokine expression. As shown in Fig 4B, following D35 administration the increase in mRNA levels for *MX1* (17.5

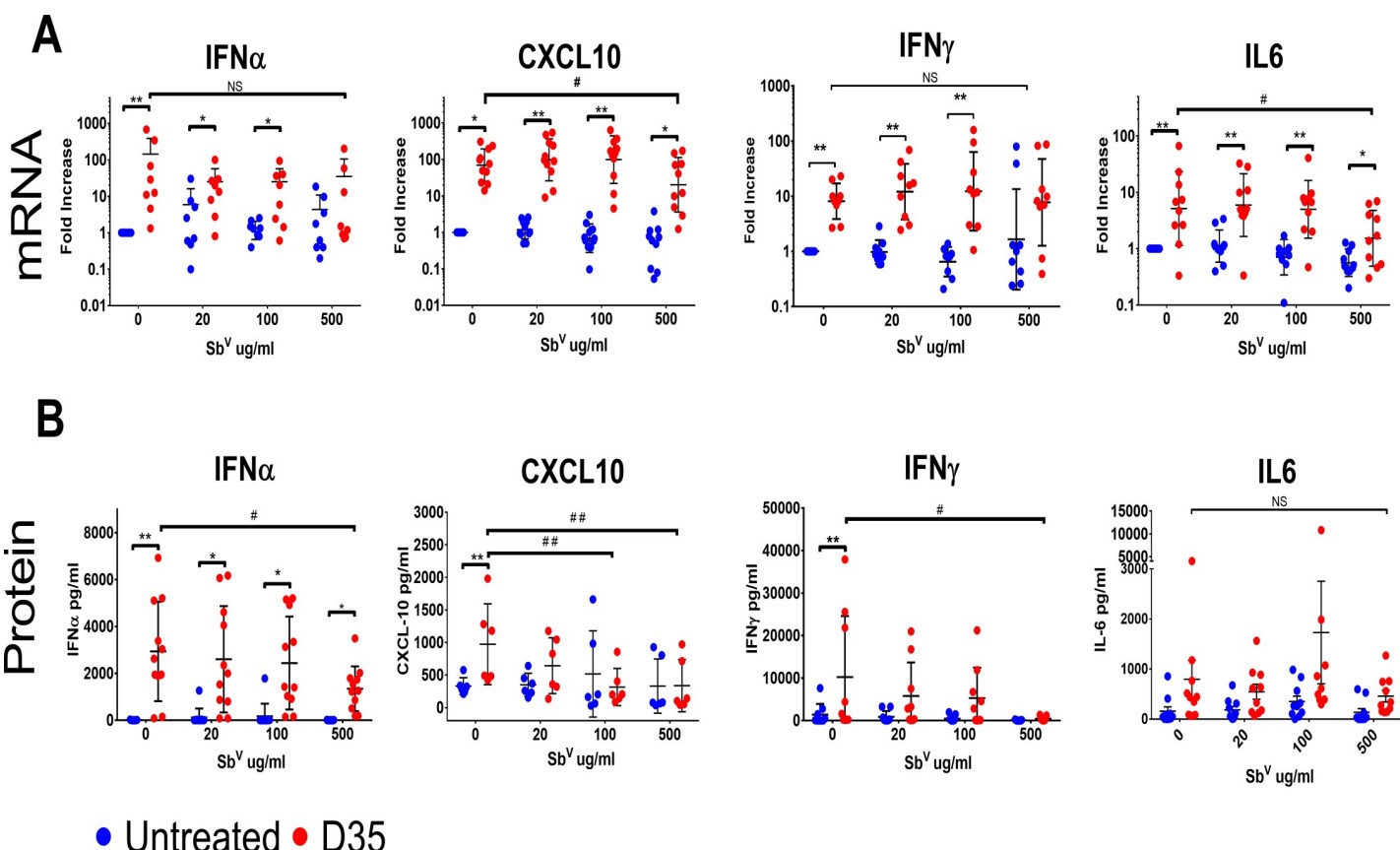

**Fig 3. Effect of Sb$^V$ on D35-induced cytokine expression *in vitro*.** Human PBMCs were treated with D35 (1μM) in the presence of increasing concentrations of Sb$^V$ (0–500 μg/ml). A) Gene expression was analyzed after 24 h of stimulation by qPCR and expressed as fold change compared to each individual donor's unstimulated sample. B) Protein levels in 72 h supernatants were determined by Luminex. For gene expression n = 10, and for protein n = 6. Mean ± SD were calculated using Graphpad Prizm. * indicated difference between untreated and D35 treatment, while # indicates significant differences between baseline and Sb$^V$ treated, where* or # for p < 0.05 and ** or ## for p<0.01. Differences in gene expression were calculated using a paired analysis.

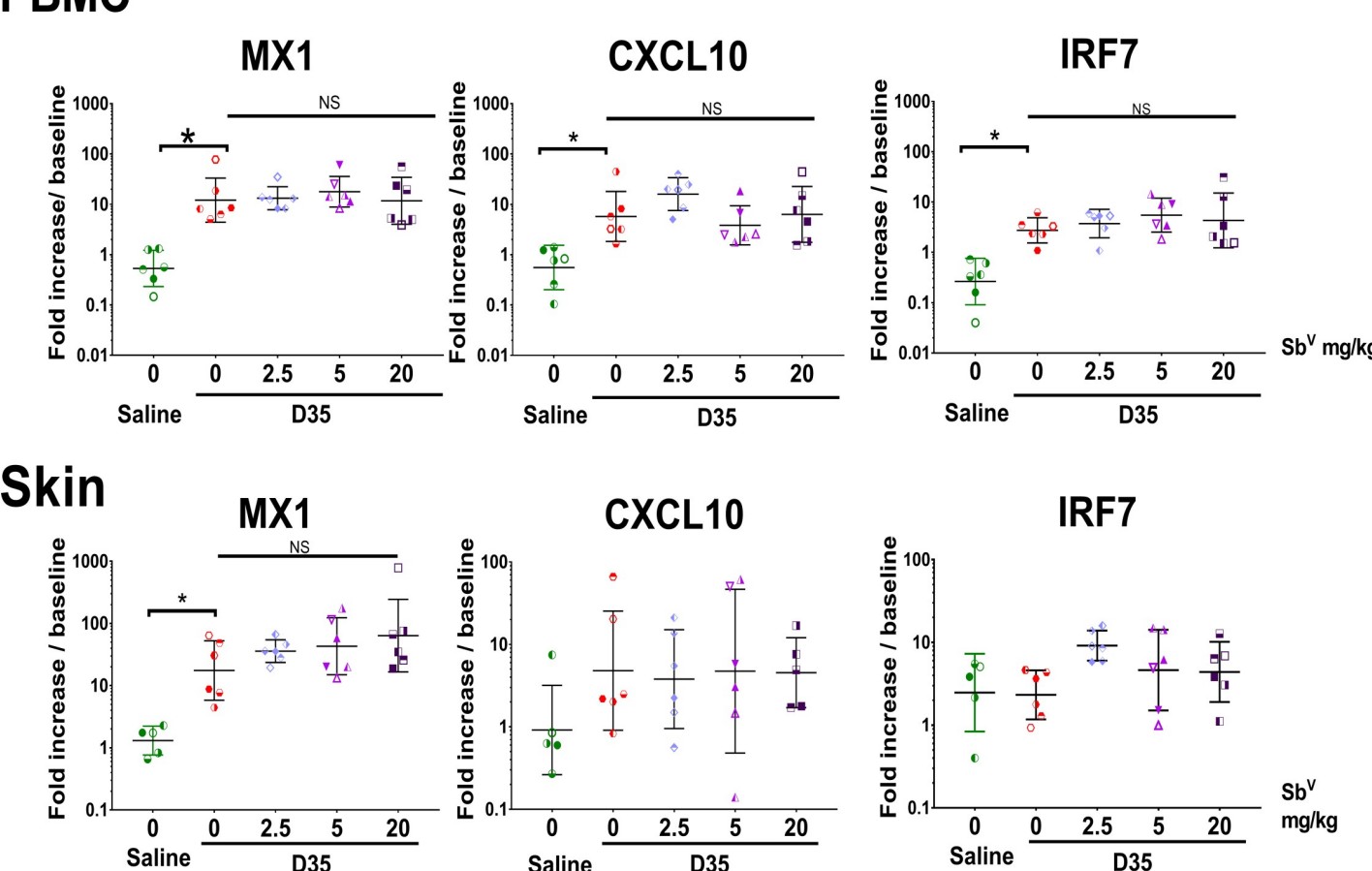

**Fig 4. $Sb^V$ effects on D35 signaling *in vivo*.** Five groups of 5 rhesus macaques were treated with $Sb^V$ at 0, 0, 2.5, 5, or 20 mg/kg/day IM for 3 days. On the 3rd day D35 (1 mg/kg SC) or saline were administered at a separate location. PBMCs and skin biopsies distant from site of drug administration were collected 24 h following the CpG ODN treatment. Cytokine mRNA levels were measured by qPCR. Individual animals represented by a unique symbol. n = 5, Geometric mean ± SD. * <0.05.

fold) and *CXCL10* (4.8 fold) in skin was quite variable between macaques, and we did not observe detectable changes in mRNA levels for *IRF7*. The D35-induced increase in mRNA expression was not modified by the $Sb^V$ treatment (Fig 4B). These data indicate that short-term systemic administration of clinically relevant doses of $Sb^V$ does not impair the effect of D35 *in vivo*, despite the modulation of IFNα and IFNγ responses to CpG ODN observed *in vitro*. Additional studies will be needed to evaluate whether prolonged exposure in vivo could impair TLR9 responses.

### Effect of $Sb^V$ and D35 treatment on gene expression in macaques with cutaneous leishmaniasis

Previous studies in macaques had established that challenges with *L. major* resulted in maculo-papular lesions that heal within 10 weeks of challenge [30]. Administration of a full course of $Sb^V$ (20mg/kg for 20 days) in infected macaques, reduced the lesion severity and accelerated healing whereas, as previously reported, a suboptimal regimen of $Sb^V$ (5 mg/kg for 10 days) did not [26] (S4 Fig). To determine whether the administration of D35 would improve the efficacy of a suboptimal regimen of $Sb^V$, 15 rhesus macaques were assigned to 4 treatment groups: Group 1 Saline, Group 2 $Sb^V$ 5 mg/kg ($Sb^V_{lo}$), Group 3 D35 1 mg/kg (D35), and Group 4 $Sb^V_{lo}$

plus D35. Macaques were challenged with 2 x 10$^6$ *L. major* metacyclic promastigotes on 3 sites on the forehead (2 sites (for biopsies) on the left forehead and 1 on the right forehead (measurement of lesion development). Following the challenge with *L. major*, the macaques were evaluated each macaque every 2–3 days for the emergence of lesions (S2 and S5 Figs). Eight days post-infection (DPI) 14 of 15 macaques had developed at least 1 lesion larger than 3 mm in diameter so the treatment phase was initiated (S5 Fig): Macaques in groups 3 and 4 were treated with a single dose of D35 SC in the hind quarters, while the macaques in groups 1 and 2 received saline (control) by the same route. At the time of D35 administration (8 DPI), there was no difference in the median lesion score between the macaques assigned to the saline or D35 groups (30±68.76 and 26.5±9.93, respectively) as measured on the right lesion site (S5 Fig). Three days later (11 DPI), the lesion score had grown for most monkeys but was relatively smaller at 41±15.97 among D35-treated animals as compared to 135±79.01 in saline treated animals (Fig 5A, p = 0.0215; S5 Fig). To determine whether we could associate differences in clinical progression with changes in local or systemic mRNA levels for immune related genes, we collected peripheral blood and a skin biopsy of 1 lesion on the left side from all animals on day 11, prior to starting Sb$^V_{lo}$ treatment. The local (skin) and systemic (whole blood) mRNA levels were assessed by NanoString analysis (770 gene panel for skin and 70 gene panel for whole blood). The mRNA expression level for each animal was standardized to its own tissue-specific gene expression level at baseline.

In whole blood, the individual differences in gene expression between treatment groups were more subtle, however, hierarchical clustering of gene expression identified a group of type I IFN-related genes in D35-treated macaques compared to infected controls (Fig 5B). Eight of the nine classical type I IFN responsive genes in the array showed significant upregulation in D35 vs saline treated rhesus macaques' whole blood (Fig 5C–5K and S6A Fig) suggesting that D35 had induced a type I interferon response. These results were confirmed by qPCR for *MX1*, *OAS1*, and *CXCL10* (S6B Fig). In addition, infected macaques treated with D35 tended to have lower levels of *IL23* and *IL17F* as well as relatively higher levels of TBET (*TBX21*), *CCL22*, granulysin, granzyme B, and perforin 1, suggesting increased levels of activated NK cells and a shift towards cytotoxic responses in peripheral blood (Fig 5J and 5L–5Q).

In untreated macaques, skin biopsies of the lesion site 11 days post challenge show differential expression of 379 of 700 genes monitored (defined as >2-fold expression change and p<0.05 relative to each monkey's baseline) as compared to 404 among those treated with D35. Importantly, while the change in expression of most genes at the lesion site was driven by the parasite and similarly modified in both groups (Fig 6A), the magnitude of the response for some genes varied by treatment (S7 Fig; for a complete list of genes altered relative to background, see S4 Table and S5 Table). Pathway analysis indicated an enrichment of paths related to innate cell function and activation including DC maturation and increased TLR receptor signaling for animals treated with D35 relative to the ones that received saline (Fig 6B). Indeed, the lesions in macaques treated with D35 had a trend towards increased levels of mRNA for chemokines that attract monocytes and neutrophils (*CCL2*, *CCL3*, *CCL5*) and adhesion molecules such as *SELE* and *SELL*, as well as a corresponding increase in genes related to APC maturation and antigen processing such as *CD80*, *CD86*, and MHC (Fig 6C and 6D). Further, levels of mRNA for genes *IL1β*, *IL6*, *C3*, and *MMP9* were higher in lesions of D35 treated macaques indicating a proinflammatory environment. While no difference was observed in T cell associated markers *CD3*, *CD4* and *CD8*, markers for NK cells *SLAMF6* and *KLR1* were increased in D35 treated macaques along with increases in *IFN*γ, granzymes B and K, and *EOMES*, but not *IL10* and *IL4*, (Fig 6E and 6F and S4 Table and S5 Table) suggesting the presence of activated NK and T cells. However, we did not observe a change in the expression of perforin or granulysin, which are also associated with the cytolytic function of NK cells and

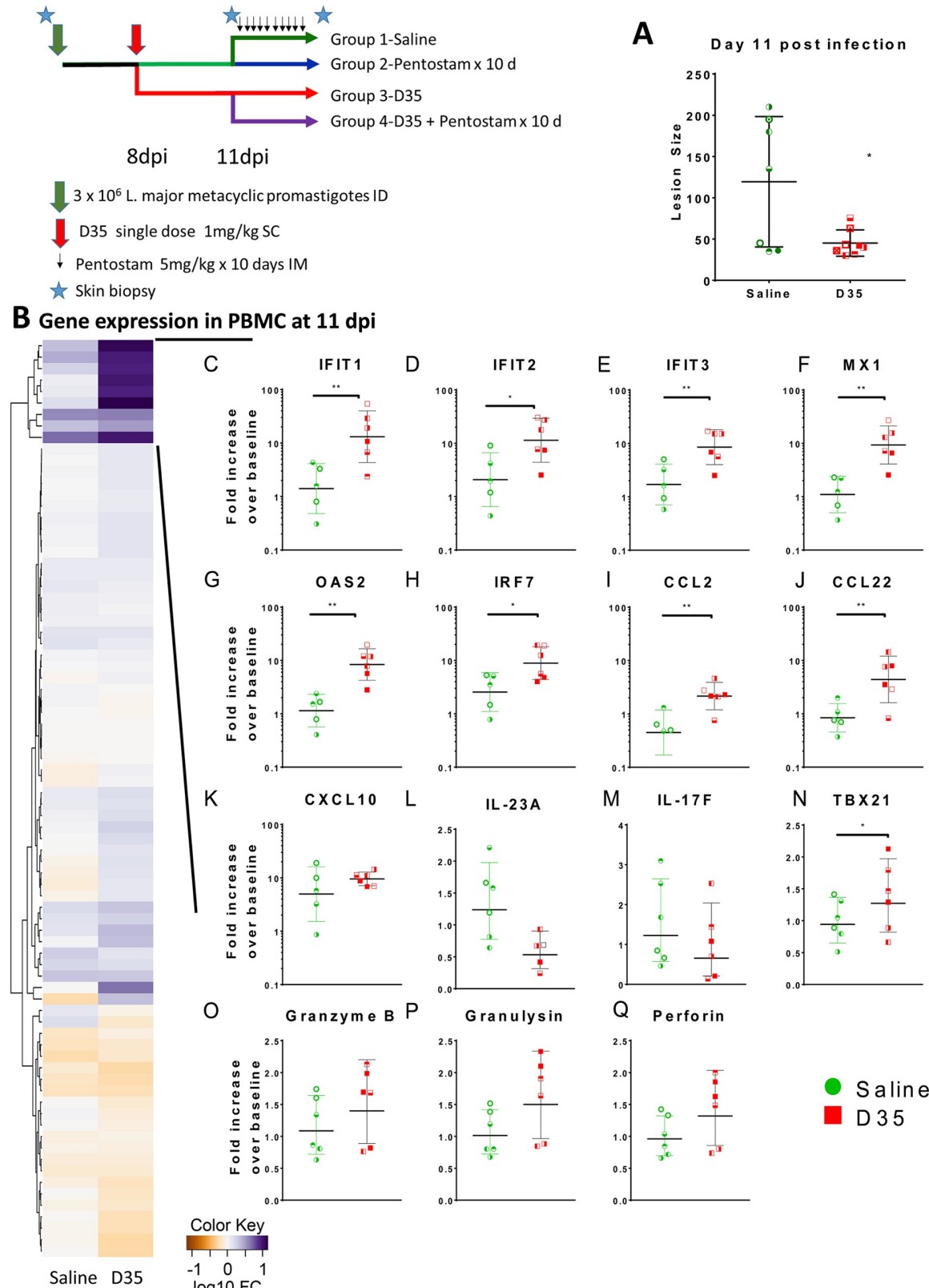

**Fig 5. Effect of D35 on gene expression in blood following an in vivo infection with *L. major*.** Fifteen rhesus macaques were challenged with 3 x10⁶ *L. major* parasites/challenge site (*L. major* metacyclic promastigotes, clone V1 promastigotes (MHOM/IL/80/

Friedlin). After 8 days, 8 macaques were treated with D35 (1 mg/kg SC). mRNA expression in peripheral blood was assessed at day 11 and compared with baseline. (A) Lesion size at D11 post-infection. (B) The expression profile of 70 genes from an immune focused panel was measured by Nanostring and the number of genes that were significantly differentially regulated compared to pre-infected blood cells were determined. (C-Q) shows the expression of a subset of genes linked to type I IFN, T cell and NK cell activation (C-K). Each animal has been given a unique symbol. n = 6, geometric mean ± SD, $^*$p < 0.05.

their ability to kill intracellular pathogens [51]. Together these data indicate that systemic administration of D35 results in a reduction in the severity of *Leishmania* lesions that is associated with increased systemic type I IFN responses and possibly the activation of NK cells and macrophages.

## D35 treatment improves lesion progression

Previous studies suggested that TLR agonists can improve the outcome in CL patients that receive standard of care treatment [20]. To assess whether D35 could improve the response to an abbreviated, low-dose course of $Sb^V$, 3 days after the administration of D35 or saline, 2 groups of macaques (saline- $Sb^V_{lo}$ or CpG ODN- $Sb^V_{lo}$) were started on a course of 5 mg/kg $Sb^V_{lo}$ (IM for 10 days). Similar low-dose antimonial courses were previously found not to be effective as single therapies in macaques challenged with *L. brazilensis* [26] or *L. major* (S4 Fig). As shown in Fig 7, macaques that received $Sb^V_{lo}$ alone had lesions comparable in maximal size to untreated macaques. Further, 1 of the 3 macaques that received 5 mg/kg of $Sb^V$ had a relapse with increased lesion severity immediately after the last dose of $Sb^V_{lo}$. In contrast, all the macaques that received a dose of D35 at 8 DPI developed milder lesions (Fig 7A). In particular, the animals that received D35 followed by $Sb^V_{lo}$ developed smaller lesions and healed earlier (Fig 7). The differences between groups were evident when comparing the maximum lesion score: 254.2±41.56 in $Sb^V_{lo}$ and 177.5±28.72 in saline-treated animals, as compared to 118.1±61.69 in D35 treated animals and 78.75±34.82 for those that received the combined treatment (Fig 7C). Accordingly, re-epithelization of the ulcerated lesion took 17.5±4 days in animals treated with D35 and 14.25±7.4 days with $Sb^V_{lo}$+D35, as compared to 28.5±7.5 days for saline and 23.7±2.31 days for $Sb^V_{lo}$-treated animals (Fig 7D). Parasite detection at the lesion site was performed using minicircle DNA qPCR at 11 or 22 DPI but failed to show significant differences in parasite loads in the lesions between groups (S8 Fig), suggesting that the reduced lesion severity is linked to a reduced inflammatory response. Alternatively, because minicircle DNA is very stable, changes in live parasites was obscured. Overall, our results demonstrate that D35 improves the outcome in macaques that receive a low dose and shortened treatment course of $Sb^V_{lo}$ and identified no negative interaction between D35 and $Sb^V$. This suggests that D35 could be used to improve the response to suboptimal regimens of $Sb^V$ reducing the risk of SAE and potentially improving their efficacy.

## Discussion

Pentavalent antimonials such as sodium stibogluconate and meglumine antimoniate are the most commonly used therapies for CL, but due to high cost, large dose, extended treatment regimes, and severe adverse events patient compliance is a major factor in disease treatment. the high dose and extended treatment regimens needed to control the parasites and cure the lesions can lead to severe adverse events and poor compliance. Further, their efficacy varies between 20 and 90%, depending on the geographic region and *Leishmania* strain and foci of resistance have been reported with increasing frequency [14]. To improve the efficacy of antimonials, several studies have tested the use of immune modulators such as imiquimod, GM-CSF, and BCG as adjuvant therapies [20, 21, 23, 52, 53]. However most of these studies

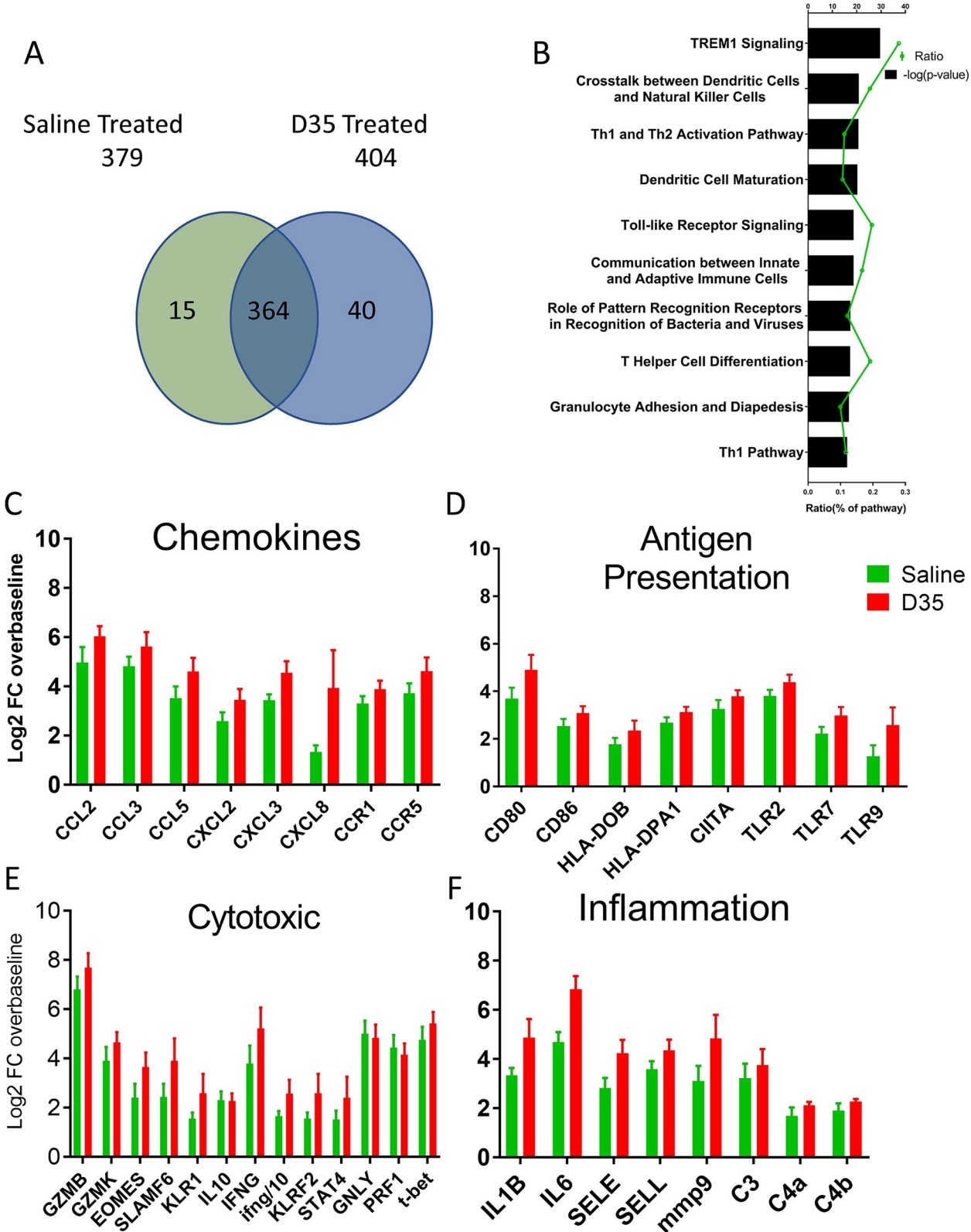

**Fig 6. Effect of D35 on gene expression in skin following an in vivo infection with *L. major*.** Fifteen rhesus macaques were challenged with 3 x $10^6$ *L. major* parasites/challenge site (*L. major* metacyclic promastigotes, clone V1 promastigotes (MHOM/IL/80/Friedlin). After 8 days, 8 macaques

were treated with D35 (1 mg/kg SC). mRNA expression in biopsies of CL lesions was assessed at day 11 and compared with baseline. The expression profile of 770 genes from an immune focused panel were measured by Nanostring and the number of genes that were significantly differentially regulated compared to pre-infected skin was identified. A) Diagram showing the distribution of genes differentially regulated ($>$2-fold change and $p<0.05$) compared to pre-infected skin. B) Ingenuity pathway analysis of gene expression in D35-treated animals vs untreated controls. Mean expression of genes related to chemokines (C), antigen presentation (D), cytotoxic genes (E), and inflammation (F) are shown. Gene expression is expressed as fold change over pre-study baseline for each individual macaque. n = 6, mean ± SEM, $^*p < 0.05$.

were aimed at improving the efficacy of full courses of antimonials (20–30 mg/kg for 3–4 weeks), but not at reducing antimonial exposure in order to improve patient safety [54]. This is partly because abbreviated and low-dose treatments with antimonials had been shown to have reduced therapeutic efficacy [15, 26]. In our study, we used D35, a TLR9 agonist that selectively activates pDC as an adjuvant treatment to $Sb^V$ in macaques challenged with *L. major* and showed that it improved the efficacy of a short course of low dose $Sb^V$ resulting in smaller lesions that re-epithelialize faster. Assessment of the local and systemic responses to the ODN showed increased expression of type I interferon-related genes as well as the upregulation of genes linked to NK cell and macrophage activation. Importantly, the innate immune activation induced was not associated with changes in temperature, behavior, CBC, liver or renal chemistries, even at doses 10-fold over the one that was therapeutically effective in macaques [15].

In the clinic, most cases of CL involve one or more painless ulcers ranging from a few mm to several cm that heal spontaneously in 3 to 18 months, although the rate varies significantly depending on the parasite strain [1]. Thus, the decision to treat is driven by the number and location of lesions, and is aimed at accelerating cure, reducing scarring and diminishing the risk of dissemination or progression to mucocutaneous leishmaniasis. Antimonials have been used to treat CL since the early 20[th] century [55] and continue to be the first line of treatment in most countries due to their low cost. In endemic countries, products such as amphotericin B are only used in patients that fail 1 or 2 courses of antimonials or for special patient populations (pregnant women, patient with renal/hepatic problems, HIV co-infected) as they are expensive, require lengthy intra-hospital administration, are associated with serious adverse events, and can foster the emergence of resistant strains [55]. Thus, to ensure accessibility, there is a need to identify treatments that are shorter, have reduced toxicity, improved compliance, reduced likelihood of emergence of resistance and reduced cost.

One possible approach is combining therapies with complementary mechanism of actions, particularly for patients with multiple CL lesions as well as patients with a disseminated form of CL or PKDL [23, 55] There are multiple studies showing that short synthetic oligonucleotides encoding one or more CpG motifs signal through TLR9 to activate the innate immune response and foster a stronger Th1 adaptive response. These have led to multiple clinical studies in infectious diseases and cancer [56, 57]. Most of those studies have used PS-ODNs (type B, K, or C), which can have strong pro-inflammatory effects and foster antibody production but were previously shown to be ineffective in CL [30, 58]. Clinical development of CpG ODN type D, which are effective in CL, has lagged behind PS-ODN due to the reduced stability during manufacture and storage. These ODN are characterized by having a core sequence with a single purine-pyrimidine-CpG-purine-pyrimidine motif flanked on both sides by 3–5 self-complementary bases on a phosphorodiester backbone capped by two phosphorothioate bases to reduce degradation. The phosphodiester backbone allows for the formation of a stem–loop conformation and/or formation of dimers [31]. In addition, D-type ODN have a 3′ end poly (G) motif, which is known to self-associate via Hoogsteen base-pairing to form parallel quadruplex structures called G-tetrads when formulated in PBS or saline [59]. While this complex conformation enables type D CpG ODN to localize to early transferrin receptor positive

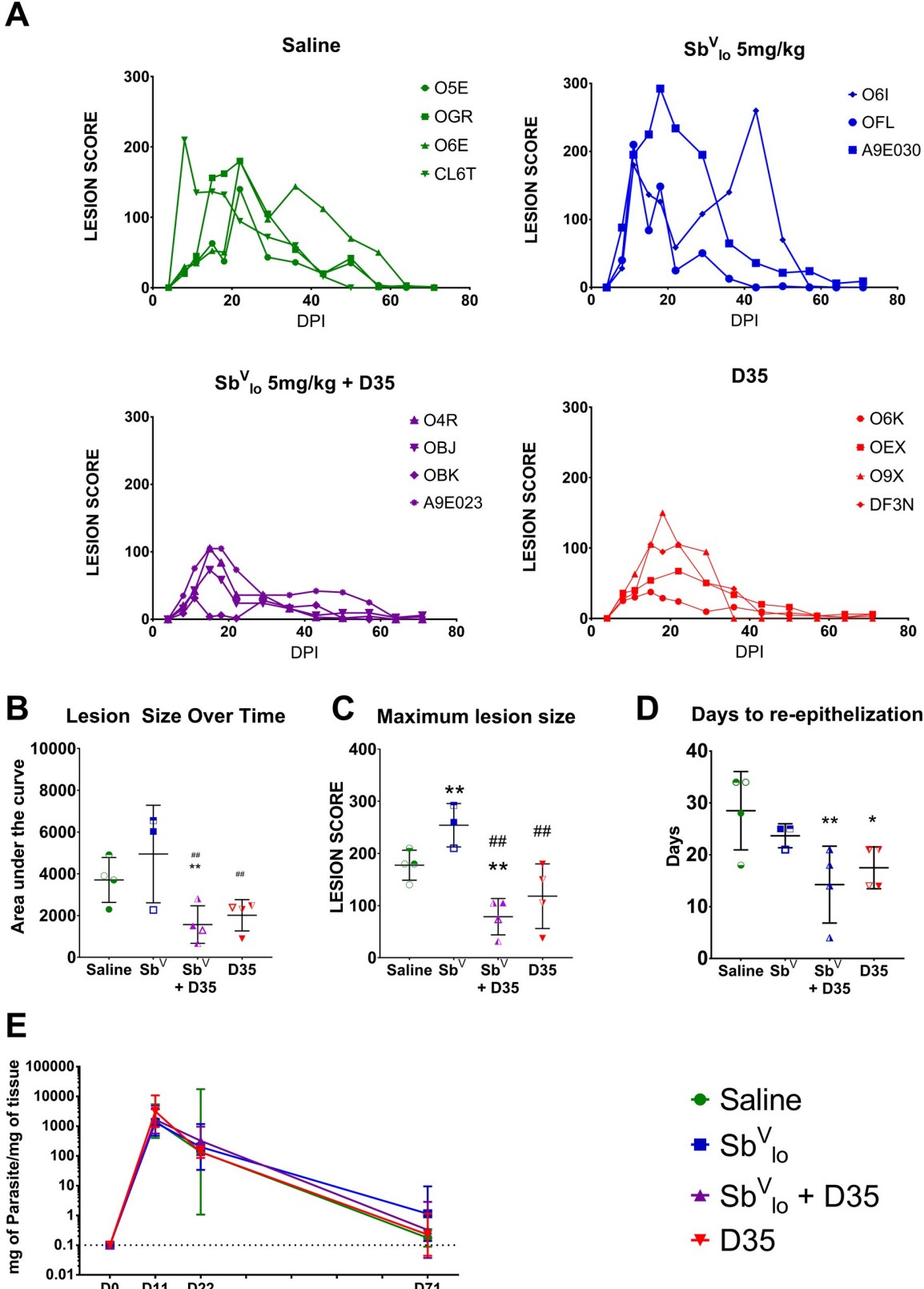

**Fig 7. D35 reduces lesion size and improves healing in response to *L. major* infection.** Fifteen rhesus macaques were challenged with 3 x 10⁶ *L. major* parasites/challenge site (*L. major* metacyclic promastigotes, clone V1 promastigotes (MHOM/IL/80/Friedlin). After 8

days, 8 macaques were treated with D35 (1 mg/kg SC). Macaques were then treated with $Sb^V$ (5mg/kg/d IM from 11–20 DPI). Lesion size was measured over the course of the study. A) Evolution of lesion development for each animal is shown by treatment group. Lesion development over time was quantified as area under the curve (B), maximum lesion size (C), and days each monkey took to re-epithelize their ulcerated lesion (D). E) Numbers of L. major parasites were quantified by qPCR using minicircle DNA isolated from a portion of a skin biopsy. Absolute numbers of parasites were calculated using a standard curve generated from biopsy samples spiked with a known number of parasites. Parasites numbers were quantified at D11, D22 and D71 post infection. Stars (*) indicate difference with saline-treated macaques. # indicates difference with $Sb^V$ treated macaques. n = 3-4/group, *$p < 0.05$, ** or ##$p < 0.01$, ***$p < 0.001$.

endosomes to signal through TLR-9 and induce IFNα, it makes the preparation unstable, posing formidable challenges for the synthesis, purification, and characterization of the oligos. Further, this can lead to significant lot to lot variation, hindering the clinical development of these ODN. To overcome these issues, we developed a formulation strategy that improves the stability of the compound and significantly reduces product impurities, with most of the product being present as monomers or dimers, which facilitates product manufacturing and characterization. As shown in Fig 1 and S1 Fig, the new formulation is stable over time and induces a similar pattern of gene expression as the research grade product when used to stimulate PBMC or a TLR9 expressing cell line *in vitro*. The reduction in large aggregates or other product-related impurities was associated with a reduction in potency at higher concentrations as determined by our *in vitro* bioactivity assay. Despite this, the formulated product induced the expression of IFNα-inducible genes in vivo in a consistent and dose-dependent manner as demonstrated by the expression of IFNα inducible genes, both at the site of administration as well as systemically in PBMC and contralateral skin biopsies. It was possible that systemic induction of type I interferons could potentially be linked to adverse effects or autoimmunity. However, our treated macaques did not show evidence of IFN-linked toxicities. This is consistent with a recent study where CpG ODN designed to activate pDC to secrete type I IFNs were used as an adjuvant therapy to a checkpoint inhibitor in patients with advanced melanoma; in these patients the main adverse effects reported consisted of local injection site injuries, rash and tenderness, and flu like symptoms (fatigue, fever, chills, myalgia) that resolved quickly following cessation of the treatment [60]. Further, unlike the ODN used in that study, D35 has only 2 phosphorothioate bases on each end, reducing the risk of non-specific inflammation, thrombocytopenia, and neutropenia that have been sporadically reported for PS ODN. Accordingly, in our current study we did not observe any negative responses from our dose escalation in cynomolgus macaques, such as liver enzymes, increased temperature, or change in any blood cell populations. Although we cannot rule out a short-term increase in temperature following administration of D35 as we only measured temperatures at 24 hours post administration, our results suggest that D35 would be well tolerated.

The mechanisms by which D35 improves the outcome of CL remain unclear. Our data shows that D35 induces a type I interferon response resulting in increased expression of multiple ISGs including MX1, IFIT1, IRF7, and CXCL10. Of note, CXCL10 was previously shown to mediate the therapeutic effect of TLR9 agonists in mice, and animals depleted of CXCL10 had a reduced CD8-T cell response to the parasites [61]. In addition, the macaques treated with D35 had increased expression of antigen presentation markers CD80, CD86, MHC-II, and transcription factor CIITA (Fig 6D). This is consistent with previous studies showing that Type D ODNs induce the rapid maturation of monocytes into DC in a type I IFN dependent manner [33] as well as with reports showing that TLR9 may mediate the IL12 in response to *L. major* in mice [62]. D35 treatment also resulted in an increase in the proinflammatory chemokines and cytokines CCL2, 3, 5, IL1β, and IL6, as well as MMP9 in the lesion, which would promote the influx and activation of monocytes (Fig 6C and 6F). The association of reduced

lesion size and increase in pro-inflammatory gene expression is interesting, particularly given that a recent study suggests that in mice, NLRP3 inflammasome activation and IL1β production could contribute to CD8 T cell mediated pathology in murine cutaneous leishmaniasis [63]. It is, however, possible that this association exists only at one end of the CL lesion spectrum, and in the context of the self-healing lesions induced by *L. major* in NHPs, a monocyte recruitment and activation and a discrete increase in pro-inflammatory cytokines may contribute to limiting the severity of the lesion. Lastly, our data on gene expression in PB and skin of *L. major* infected-D35 treated macaques suggests that D35 induced higher levels of multiple genes associated with NK cell activation at the lesion site than were activated by the infection alone. Of note, the pattern of gene expression is not complete, as there are increased levels of granzymes B and K, SLAMF6, KLR1, as well as T-bet and Eomes, but not perforin 1 or granulysin. Human NK cell response to Leishmania is still poorly characterized, but a recent study from Caneda-Guzman describes different degrees of NK cell activation in the lesion of patients with localized and diffuse CL (*L. Mexicana*) and proposes that NK cells play a key role in defining the severity of the lesions [64]. Lastly, D35 could increase parasite clearance by activating macrophages or neutrophils. Surprisingly, in our study, the significant reduction in lesion size in animals treated with D35 alone or D35 plus $Sb^V$ was not associated with changes in parasite numbers at any point during the study. These results are different from previous studies [30, 65]. It is possible that this was the result of using minicircle DNA as a readout for parasite load. While this method is very sensitive and kDNA is highly abundant (10,000–20,000 copies) and stable, the method is not optimal for discriminating between live and dead parasites, and remnants of DNA may have masked differences between groups. Indeed, it was only after the study was completed at 71 DPI, that we observed a significant decrease in parasite numbers by qPCR. Additional studies will need to be done to confirm the effect of D35 on parasite load; however, multiple studies have shown that the number of parasites in a lesion does not always correspond to its severity [55, 65, 66]. At this time, it is not possible to determine the exact mechanism by which D35 improves the outcome in the CL model, but the data show that it induces significant immune responses *in vivo* and modifies the response to the parasite.

The mechanism of action for $Sb^V$ is also not clear despite it being used for almost 100 years [55, 67]. This is partly because $Sb^V$ consists of a mixture of oligomeric structures. $Sb^V$ could behave as a prodrug being reduced *in vivo* to a more active/toxic trivalent form of antimony (Sb(III)) that exhibits antileishmanial activity through inhibition of trypanothione reductase [55, 67]. Of note, it is not clear whether this reduction occurs in the parasite, the macrophages or both. In addition, $Sb^V$ could act as an inhibitor of type I DNA topoisomerase leading to reduced ATP (adenosine triphosphate) and GTP (guanosine triphosphate) synthesis. Lastly, more recent studies suggest that $Sb^V$ could enhance parasite killing by macrophages through an increase in TNFα, ROS, and NO levels. Interestingly, studies in mice suggest that Th1 responses enhance the leishmanicidal effect of $Sb^V$[55]. It is known that HIV-infected patients, who have low levels of T cells, respond poorly to $Sb^V$. Therefore, it is possible that D35 enhances the effect of a low-dose abbreviated course of $Sb^V$ by increasing the levels of NO, ROS and Th1, enhancing the leishmanicidal effect of $Sb^V$ while modulating the inflammation of the site and reducing the severity of the lesions. Regardless of the underlying mechanism of action, our study suggests that the use of D35 as an adjuvant therapy can improve the efficacy of low dose $Sb^V$ treatment thereby reducing the patient's exposure and diminishing the risk of adverse effects. This approach could result in improved patient compliance and accelerated healing, helping curve the emergence of resistance. The results presented here argue for the effectiveness of D35 in combination with $Sb^V$, and future clinical trials in humans are warranted to further explore D35 as an adjunct therapy to low dose $Sb^V$ treatment. Interestingly, CpG ODN type D were shown to reduce the severity of CL lesions in SIV infected macaques.

Patients with HIV taking ART have the highest $Sb^V$-associated toxicities [55, 67, 68]; future studies will need to determine whether the addition of D35 as an adjuvant therapy can improve the clinical outcome in these patients. In summary, our data suggests exploring the clinical use of field-friendly, affordable, and apparently safe synthetic oligonucleotide D35 in combination with the current antimonial treatments to reduce the dose, duration and side effects of current therapies, and increase compliance is warranted.

## Supporting information

**S1 Methods.**
(DOCX)

**S1 Fig. Design of dose escalation study in Cynomolgus Macaques.**
(TIF)

**S2 Fig. Graphical representation *L. major* infection and treatment study.**
(TIF)

**S3 Fig. Physical characterization of ultrapure and research grade D35.**
(TIF)

**S4 Fig. Low dose $Sb^V$ results in recrudescence.**
(TIF)

**S5 Fig. Evolution of *L. major* lesions.**
(TIF)

**S6 Fig. Detection of gene signature in PMBCs of macaques treated with D35.**
(TIF)

**S7 Fig. Gene signature in skin of macaques treated with D35.**
(TIF)

**S8 Fig. Leishmania quantification by minicircle qPCR.**
(TIF)

**S1 Table. Genes in Custom NanoString code set.**
(DOCX)

**S2 Table. Ingenuity Canonical Pathways.**
(DOCX)

**S3 Table. Clinical chemistry values and temperatures for cynomolgus macaques.**
(DOCX)

**S4 Table. Genes 2-fold increased over baseline saline.**
(DOCX)

**S5 Table. Genes 2-fold increase over baseline D35.**
(DOCX)

## Acknowledgments

The authors thank Dr. Amy Rosenberg, Ashutosh Rao and Hira Nakhasi for reviewing the manuscript. Dr T. Maeda kindly provided the CAL-1 cells, Dr David Sacks prepared and provided the MHOM/IL/80/Friedlin *L. major* clone VI promastigotes, and Dr Hideaki Sato from

GeneDesign provided the ultrapure D35. In addition, we thank Dr. John Dennis, Jill Ascher, Lewis Shankle, and the Animal Care Facility staff for their care of the non-human primates included in this study. Ian L. McWilliams and Swaksha Rachuri are fellows at the Postgraduate Research Participation Program at the Center for Drug Evaluation and Research administered by the Oak Ridge Institute for Science and Education through an interagency agreement between the U.S. Department of Energy and the U.S. Food and Drug Administration. We thank Yolanda Hawkins and the FDA Office of Translational Science and technology transfer supporting the collaboration. We thank the Dr Analia Porras, PAHO, and the WHO Demonstration Project Initiative for their support. The assertions herein are the private ones of the authors and are not to be construed as official or as reflecting the views of the Food and Drug Administration at large.

## Author Contributions

**Conceptualization:** Lydia Halie, Serge Beaucage, Robert Duncan, Farrokh Modabber, Graeme Bilbe, Byron Arana, Daniela Verthelyi.

**Funding acquisition:** Beatrice Bonnet.

**Investigation:** Seth G Thacker, Ian L. McWilliams, Lydia Halie, Swaksha Rachuri, Ranadhir Dey.

**Methodology:** Seth G Thacker, Ranadhir Dey, Daniela Verthelyi.

**Project administration:** Beatrice Bonnet.

**Resources:** Stephen Robinson.

**Supervision:** Daniela Verthelyi.

**Writing – original draft:** Seth G Thacker, Daniela Verthelyi.

**Writing – review & editing:** Seth G Thacker, Beatrice Bonnet, Serge Beaucage, Robert Duncan, Farrokh Modabber, Graeme Bilbe, Byron Arana, Daniela Verthelyi.

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
