## [Decision Letter · Decision Letter 0]

11 Nov 2019

Dear Dr Verthelyi:

Thank you very much for submitting your manuscript "CpG ODN D35 improves the response to abbreviated low-dose pentavalent antimonial treatment in non-human primate model of cutaneous leishmaniasis" (#PNTD-D-19-01411) for review by PLOS Neglected Tropical Diseases. Your manuscript was fully evaluated at the editorial level and by independent peer reviewers. The reviewers appreciated the attention to an important problem, but raised some substantial concerns about the manuscript as it currently stands. These issues must be addressed before we would be willing to consider a revised version of your study. We cannot, of course, promise publication at that time.

We therefore ask you to modify the manuscript according to the review recommendations before we can consider your manuscript for acceptance. Your revisions should address the specific points made by each reviewer. 

When you are ready to resubmit, please be prepared to upload the following:

(1) A letter containing a detailed list of your responses to the review comments and a description of the changes you have made in the manuscript.

(2) Two versions of the manuscript: one with either highlights or tracked changes denoting where the text has been changed (uploaded as a "Revised Article with Changes Highlighted" file); the other a clean version (uploaded as the article file).

(3) If available, a striking still image (a new image if one is available or an existing one from within your manuscript). If your manuscript is accepted for publication, this image may be featured on our website. Images should ideally be high resolution, eye-catching, single panel images; where one is available, please use 'add file' at the time of resubmission and select 'striking image' as the file type. 

Please provide a short caption, including credits, uploaded as a separate "Other" file. If your image is from someone other than yourself, please ensure that the artist has read and agreed to the terms and conditions of the Creative Commons Attribution License at http://journals.plos.org/plosntds/s/content-license (NOTE: we cannot publish copyrighted images). 

(4) If applicable, we encourage you to add a list of accession numbers/ID numbers for genes and proteins mentioned in the text (these should be listed as a paragraph at the end of the manuscript). You can supply accession numbers for any database, so long as the database is publicly accessible and stable. Examples include LocusLink and SwissProt.

(5) To enhance the reproducibility of your results, we recommend that you deposit your laboratory protocols in protocols.io, where a protocol can be assigned its own identifier (DOI) such that it can be cited independently in the future. For instructions see http://journals.plos.org/plosntds/s/submission-guidelines#loc-methods

While revising your submission, please upload your figure files to the Preflight Analysis and Conversion Engine (PACE) digital diagnostic tool, https://pacev2.apexcovantage.com/ PACE helps ensure that figures meet PLOS requirements. To use PACE, you must first register as a user. Then, login and navigate to the UPLOAD tab, where you will find detailed instructions on how to use the tool. If you encounter any issues or have any questions when using PACE, please email us at figures@plos.org.

We hope to receive your revised manuscript by Jan 10 2020 11:59PM. If you anticipate any delay in its return, we ask that you let us know the expected resubmission date by replying to this email.

To submit a revision, go to https://www.editorialmanager.com/pntd/ and log in as an Author. You will see a menu item call Submission Needing Revision. You will find your submission record there. 

Sincerely,

Abhay R Satoskar

Associate Editor

Steven Singer

Deputy Editor

Reviewer's Responses to Questions

**Key Review Criteria Required for Acceptance?**

**Methods**

-Are the objectives of the study clearly articulated with a clear testable hypothesis stated?

-Is the study design appropriate to address the stated objectives?

-Is the population clearly described and appropriate for the hypothesis being tested?

-Is the sample size sufficient to ensure adequate power to address the hypothesis being tested?

-Were correct statistical analysis used to support conclusions?

-Are there concerns about ethical or regulatory requirements being met?

Reviewer #1: The paper presented by Thacker et al is about the efficacy of one dose CpG ODN D35 with low dose antimonial treatment in L. major infected primates. Their results showed that by using this approach, macaques had smaller lesions and reduced time to re-epithelization but almost no effect on parasite load. 

The manuscript is very important in the field of CL treatment especially the data presented in macaques. The main problem in this manuscript is incorporation of too many data which indeed very hard to follow. Especially the graphs are not prepared properly and are very hard to read due to used small letters on graph heading.

M@M section

Line 151, "Buffy coats from healthy blood donors" Please mention healthy human blood donor?

Line 163-169: why for CAL‐ 1 cells, the combination of D35+ SbV did not use? Is there specific reason for time variation sometimes it is 24 hr for PBMC and 18hr for CAL‐ 1 cells ?

Lines 197-200: please clarify this sentence: "CpG ODN injections were performed on alternating sides of the thorax and at least 2 inches away from a previous inoculation site"

Line231: "promastigotes on the forehead at three separate sites (1 on the right and 2 on the left)"what is the reason? Why the total parasites did not injected at one point? 

Line 229: Please define the treatment groups as groups 1 , 2, 3, 4 similar to the text 458-459.

Line 238: please define DPI

Reviewer #2: Methods are comprehensive and adequate

**Results**

-Does the analysis presented match the analysis plan?

-Are the results clearly and completely presented?

-Are the figures (Tables, Images) of sufficient quality for clarity?

Reviewer #1: Lines 305 -332: why the authors did not use macaques instead of human PBMC? What was the limitation? Please harmonize the term D35 or D ODN or ODN….it is better to use D35 throughout the manuscript. Define MX-1 in Fig 1D and E.

Line 307: please change to L. major

Line 324: what about IFN-g in Fig 1C?

Line 324: Fig 1D needs more explanation. 

Line 329-332: Please make it clear which D35 did you use for whole study in primate, ultrapure or Research grade?

Fig 2 A-D: in some part the control macaque is missing (purple line), it is really hard to understand this figure very well. In this section D35 is used every 3 weeks for 6 times? Please clarify.

Fig 2 G-L: what is the orange line? G11 shown with purple line which is only appear on first injection period! Very hard to understand.

Line 353: What is the reason to have considerable expression of interferon stimulated genes (ISGs), MX-1, OAS-1, CXCL-10, and IRF7, in contralateral side of the torso? 

There is no graph to show the ISGs in Figure 2. In Fig 2-C there is no control at week 18. 

In Fig 2 B in contralateral side, at week 12 and 18 there is confused up and down expression for OAS1, what are the reasons?

In Fig 2 E and F, for primate C07, there is upper level of IL-6 and IL-8 in contralateral side. Altogether this graph shows lots of variation and there is no concrete conclusion. This variation is within 3 macaques. Instead of graphs, could be easier to show them in a table (mean+/- SD) with their p value. 

Line 387: this section is on human PBMC and the action of Sb and D35 at in vitro. In order not to back and forth from human to macaques, highly suggest describing everything in human first and then continued with in vitro and in vivo experiments in macaques. 

Line 398, The data for ISG is missing? 

Why the n is different for gene and protein evaluation? In Fig 3 mentioned untreated marked as circle blue and D35 as red cube. Please define the meaning of each clearly. In Fig 3A and B, why there is different view in the case of IL-6? There is no IL-6 production at protein level.

Line 418: In section "In vivo effects of SbV on D35 treatment" the authors used different gene expression for macaques (MX-1, CXCL-10 and IRF-7) and in the case of human IFN-a, CXCL10, IFN-g and IL-6, why is that so? 

Line 453: please indicate as group 1 (saline controls), group 2( SbV 5 mg/kg (SbV

lo), group 3 D35 1 mg/kg (D35), and group 4 (Sb lo plus D35). The Sup Fig 3 A is confusing and there are different flashes below the line with no meaning. 

Line 492: the Fig 5, J is missing, should be Fig5J, L-Q.

 Fig 6 D, please harmonize the y axis and all should be fixed at 10.

My major critic is in Supplementary Fig 6. This Figure is very important and should be among the essential figure not in supplementary. If there is an improvement in the lesion size, why there is no significant difference between Sb+D35 and saline or other groups. Even at D71 almost no parasite DNA was seen on saline and D35 but existed on Sb and Sb+D35. If the minicircle DNA is very stable why there are huge amount of differences between D22 and D71. It is highly suggest repeating the parasite detection with different set of primers such as RV1 and RV1 and check the differences on D11, D22 and D71.

Reviewer #2: yes

**Conclusions**

-Are the conclusions supported by the data presented?

-Are the limitations of analysis clearly described?

-Do the authors discuss how these data can be helpful to advance our understanding of the topic under study?

-Is public health relevance addressed?

Reviewer #1: Lines 568-570: need revision

Lines 668-680: it is better to use another set of primers for parasite burden and then conclude this section. This is hard to believe unless prove it by another approach

The rest of discussion is clear and reads well. It seems that different sections are written by different co-authors so some parts are written very well and some parts are very hard to understand. Therefore highly suggest improving and clarifying the result and M&M sections for reading.

Reviewer #2: yes

**Editorial and Data Presentation Modifications?**

Reviewer #1: Graphs head lines used small letters, so highly suggest to modify them.

Parasite detection is major problem. This test is highly important for their conclusion. Highly suggest to check with another set of primers such as RV1 and RV2 and compare their results with this approach.

Reviewer #2: minor revision

**Summary and General Comments**

Reviewer #1: The manuscript is very important in the field of CL treatment especially the data presented in macaques. The main problem in this manuscript is incorporation of too many data which indeed very hard to follow. Especially the graphs are not prepared properly and are very hard to read due to used small letters on graph heading.

My major critic is in Supplementary Fig 6. This Figure is very important and should be among the essential figure not in supplementary. If there is an improvement in the lesion size, why there is no significant difference between Sb+D35 and saline or other groups. Even at D71 almost no parasite DNA was seen on saline and D35 but existed on Sb and Sb+D35. If the minicircle DNA is very stable why there are huge amount of differences between D22 and D71. It is highly suggest repeating the parasite detection with different set of primers such as RV1 and RV1 and check the differences on D11, D22 and D71.

Reviewer #2: This manuscript provides pre-clinical non-human primate evaluation of combination of new drug(adjvant) namely D35 CpG ODN and low-dose current anti-Lesihmaniasis drug (antimonials) against a neglected disease cutaneous leishmaniasis. It is a very important study as cutaneous leishmaniasis is re-emerging in the unsettled parts of the world.

Based on and following their earlier studies, they evaluated several novel applications of a novel therapy modality which can be summarized into two:

1. They evaluated doses and safety of clinical grade D35 CpG ODN to be used in further studies in humans. They overcome preparation of reproducible D35 CpG ODN production and showed its safe administration sc.

2. Further, they evaluated bi-modal interaction between D35 CpG ODN (a single dose sc) and low-dose antimonials. In the comprehensive experiments they concluded that low-dose and therefore safer antimonials and D35 CpG ODN are synergistically reducing the lesion sizes and duration. 

Overall, the study is performed with delicate and enormous experimental design and written very well. 

Specific comments:

1. It should be better to make a specific cartoon-like scheme (like in Supp Figure 3A) in each in vivo study (that what was done and when treated and when samples collected), to ease readers’ understanding. 

2. Authors checked parasite DNA in the lesions during the course of infection and did not find any difference between treatment groups. I think later time point of analysis also required. 

3. I am curious what would be the cost of 1 dose D35 CpG ODN and low dose antimonials in an adult human?

4. Is there any potential of this new treatment regimen to be able to use for other Lesihmania spp?

Minor comments:

1. Line 144: “a lot” is not a scientific terminology, please erase.

2. Line 216: “where” should be “were”.

3. Line 307: Leishmania major should be written with upper letter “L”.

4. Line 484: “node” should be “role”.

5. Line 592: “and limited efficacy” needs to be erased.

PLOS authors have the option to publish the peer review history of their article (what does this mean?). If published, this will include your full peer review and any attached files.

Reviewer #1: Yes: Sima Rafati, Ph.D Dept. of Immunotherapy and Leishmania Vaccine Research, Pasteur Institute of Iran

Reviewer #2: No

---

## [Editor Report · Decision Letter 1]

12 Jan 2020

Dear Dr Verthelyi,

We are pleased to inform you that your manuscript, "CpG ODN D35 improves the response to abbreviated low-dose pentavalent antimonial treatment in non-human primate model of cutaneous leishmaniasis", has been editorially accepted for publication at PLOS Neglected Tropical Diseases.

Before your manuscript can be formally accepted and sent to production you will need to complete our formatting changes, which you will receive in a follow up email. Please note: your manuscript will not be scheduled for publication until you have made the required changes.

IMPORTANT NOTES

* Copyediting and Author Proofs: To ensure prompt publication, your manuscript will NOT be subject to detailed copyediting and you will NOT receive a typeset proof for review. The corresponding author will have one final opportunity to correct any errors when sent the requests mentioned above. Please review this version of your manuscript for any errors.

* If you or your institution will be preparing press materials for this manuscript, please inform our press team in advance at plosntds@plos.org. If you need to know your paper's publication date for media purposes, you must coordinate with our press team, and your manuscript will remain under a strict press embargo until the publication date and time. PLOS NTDs may choose to issue a press release for your article. If there is anything that the journal should know, please get in touch.

*Now that your manuscript has been provisionally accepted, please log into EM and update your profile. Go to http://www.editorialmanager.com/pntd, log in, and click on the "Update My Information" link at the top of the page. Please update your user information to ensure an efficient production and billing process.

*Note to LaTeX users only - Our staff will ask you to upload a TEX file in addition to the PDF before the paper can be sent to typesetting, so please carefully review our Latex Guidelines [http://www.plosntds.org/static/latexGuidelines.action] in the meantime.

Best regards,

Abhay R Satoskar

Associate Editor

Steven Singer

Deputy Editor

---

## [Editor Report · Acceptance letter]

25 Feb 2020

Dear Dr Verthelyi,

We are delighted to inform you that your manuscript, "CpG ODN D35 improves the response to abbreviated low-dose pentavalent antimonial treatment in non-human primate model of cutaneous leishmaniasis," has been formally accepted for publication in PLOS Neglected Tropical Diseases.

Best regards,

Serap Aksoy

Editor-in-Chief

Shaden Kamhawi

Editor-in-Chief
